# Image Quality Analysis of Photon-Counting CT Compared with Dual-Source CT: A Phantom Study for Chest CT Examinations

**DOI:** 10.3390/diagnostics13071325

**Published:** 2023-04-03

**Authors:** Marine Deleu, Jean-Baptiste Maurice, Laura Devos, Martine Remy, François Dubus

**Affiliations:** 1Medical Physics Department, University Hospital, 59037 Lille, France; 2Radiology Department, Heart-Lung Institute, University Hospital, 59037 Lille, France

**Keywords:** photon counting, iterative reconstruction algorithm, noise power spectrum, task transfer function, detectability index

## Abstract

A comparison was made between the image quality of a photon-counting CT (PCCT) and a dual-source CT (DSCT). The evaluation of image quality was performed using a Catphan CT phantom, and the physical metrics, such as the noise power spectrum and task transfer function, were measured for both PCCT and DSCT at three CT dose indices (1, 5 and 10 mGy). Polyenergetic and virtual monoenergetic reconstructions were used to evaluate the performance differences by simulating a Gaussian spot with a radius of 5 mm and calculating the detectability index. The highest iterative reconstruction level was able to decrease the noise by about 70% compared with the filtered back projection using a parenchyma reconstruction kernel. The PCCT task transfer functions remained constant, while those of the DSCT increased with the reconstruction strength level. At monoenergetic 70 keV, a 50% decrease in noise was observed for DSCT with image smoothing, while PCCT had the same 50% decrease in noise without any smoothing. The PCCT detectability index at a reconstruction strength level of two was equivalent to the highest level of ADMIRE 5 for DSCT. The PCCT showed its superiority over the DSCT, especially for lung nodule detection.

## 1. Introduction

Computed tomography (CT) technology has advanced in the past decades, improving image quality while lowering patient radiation exposure [1]. Current CT scanners have evolved into spectral CT, such as dual-source spectral CT (DSCT), which was released by Siemens in 2006 [2], as well as dual-layer CT scanners and fast kV switching CT scanners, released by Philips and GE or Toshiba, respectively [3]. Siemens dual-source CT scanners yield spectral information and use two different voltages on each X-ray tube, with an associated beam filtration to yield the spectral information. Spectral CT scanners are then based on solid-state scintillation detectors, such as current CT scanners. Absorbed X-rays are converted by the scintillator into visible light photons, which are then converted into electrons with the help of a backside photodiode. The created electrical current amplitude is then proportional to the energy of the absorbed photon. The main drawback of this detector technology is that low-energy photons contribute less than high-energy photons. Furthermore, image noise and CT number drifts are caused by electronic noise at low X-ray flux. The DSCT detection used in this study is based on Stellar^Infinity^ detectors, which are third-generation detectors with reduced electronic noise, even in the case of low tube intensity [4].

Recently, photon-counting detectors have been implemented in CT, which may become state-of-the-art CT detection technology in a few years. A commercial version of dual-source photon-counting CT (PCCT) was released by Siemens in 2021 [5,6]. Photon-counting detectors are based on the direct conversion of X-ray photons using a semiconductor material [7] made of cadmium telluride (CdTe). The voltage pulse height is proportional to the energy of the absorbed photon. When the lowest threshold energy is overcome, the X-ray photon is counted, and its energy is classified by means of three other thresholds. Hence, PCCT scanners enable spectral acquisition using a single detector. Electronic noise is eliminated by the low threshold energy (lower than 20 keV), which is not the case for standard indirect conversion detectors [8]. The main challenge for this new detection technology is in achieving an acceptable detector layer thickness in the range of millimeters. Therefore, better geometric efficiency is provided by a thinner detector layer and smaller pixel elements compared to scintillator detectors. Therefore, a high spatial resolution is reached, and pile-up effects are reduced. A PCCT ultra-high spatial resolution (UHR) is achieved with the QuantaMax detector elements, which have an area of 0.150 × 0.176 mm^2^ given at the isocenter. The standard acquisition mode means that pixels are grouped, in pairs, in each direction into a macropixel.

Since the first clinical PCCT was released, several technical and clinical studies have been performed and have shown that image quality was improved while keeping the dose constant [9,10,11,12,13,14,15,16]. As the PCCT is used in our thoracic imaging department, the goal of our study was to investigate the PCCT performance in lung nodule detection using physical metrics, which were compared with those obtained using DSCT, which was previously used in the same department. Noise power spectrum (NPS), task transfer function (TTF) and detectability index (d’) were evaluated within the Catphan 600 phantom dedicated to imaging. Firstly, the physical metrics were measured for both CT scanners using polyenergetic reconstruction since the parenchyma reconstruction kernel is used in clinical routine. Secondly, these metrics were evaluated using a virtual monoenergetic reconstruction at 70 keV. The consistency of the new iterative reconstruction (IR) algorithm linked to the PCCT was investigated by changing the reconstruction strength level and the volume computed tomography dose index (CTDI_vol_).

## 2. Materials and Methods

### 2.1. Acquisition and Reconstruction of the Phantom

The Catphan 600 phantom (Phantom Laboratory, Salem, MA, USA) is specifically designed for quality control in CT imaging. The CTP489 module was used to measure NPS, and the CTP404 module was used to determine the TTF of an air insert.

The phantom was scanned with a 144-sliced Naeotom Alpha PCCT scanner, software version VA40A (Siemens Healthcare GmbH, Erlangen, Germany), and with a 192-sliced Somatom Force DSCT scanner, software version VB20A (Siemens Healthcare GmbH, Erlangen, Germany). The maximal collimations in the cranio-caudal direction were 144 × 0.4 mm and 192 × 0.6 mm at the isocenter for PCCT and DSCT, respectively. For the DSCT scanner, the maximal fields of view (FOV) of the two detectors are 50 cm and 26 cm, while the second detector has a maximal FOV of 36 cm for the PCCT scanner.

Acquisitions on both CTs were based on a thoracic protocol at different CTDI_vol_: 1, 5 and 10 mGy, which correspond to a low-dose examination, an intermediate-dose examination in our department and a higher-dose examination, corresponding to the French national diagnostic reference level (DRL) [17]. By adapting the mAs, CTDI_vol_ display values equivalent to the wanted CTDI_vol_ were reached. Both systems’ CTDI_vol_ display values were controlled before the measurements and were in conformity with French national regulations [18].

The slice thickness was fixed at 1 mm for each phantom acquisition. The other acquisition and reconstruction parameters are listed in Table 1. For the polyenergetic reconstruction, the reconstruction kernels Bl56 and Bl57 dedicated to parenchyma were used for PCCT and DSCT, respectively. Bl56 and Bl57 were used because they are the closest kernels available on the respective CT, as the same kernel was not available. The kVp 80/Sn150 couple was chosen to correspond to the clinical routine for thorax examination on DSCT. For the virtual monoenergetic reconstruction at 70 keV, the Qr40 kernel was applied for both CT scanners to obtain virtual monoenergetic images (VMIs). The 70 keV energy was chosen because CT acquisitions without iodine injection have been reconstructed at this energy in clinical routine. Both DSCT X-ray tubes and their associated detectors were used during acquisition, while only the X-ray tube with a maximal FOV of 50 cm was needed to perform the PCCT virtual monoenergetic reconstructions.

Table 1 shows that IR algorithms and all the available reconstruction strength levels were applied to each CT acquisition. When no iteration was performed, the reconstruction was equivalent to a weighted filtered back projection (WFBP). For the DSCT scanner, the iterative algorithm Advanced Modeled Iterative Reconstruction (ADMIRE) was applied at each iteration level from one to five. Its model is based on the IR algorithm that uses advanced noise reduction methods such as noise and system modeling [16]. The PCCT scanner was released with a new iterative algorithm, namely, Quantum IR (QIR), with four strength levels. Statistical optimization of spectral data to reduce target noise level and a correction for geometric cone beam artifacts were included by default [10,12]. The higher the iteration level, the smaller the noise.

### 2.2. Physical Metrics

Prior to physical metrics evaluation, the CT numbers within the air insert and background of each CT acquisition of the phantom were measured to estimate the contrast, as recommended in the AAPM report [19].

The software imQuest v7.1 (Clinical Imaging Physics Group, Department of Radiology, Duke Health, imquest@duke.edu) was chosen to evaluate the NPS, TTF and detectability index. The calculation is based on the AAPM report 266, which is dedicated to CT performance [19]. The details and formulas of the NPS, TTF and detectability index can also be found in the AAPM report [19].

#### 2.2.1. Noise Power Spectrum

The one-dimensional NPS was measured in the homogeneous CTP489 module using five square regions of interest of 3.8 × 3.8 mm^2^, as shown in Figure 1. More than 90 slices were used for the calculation. Furthermore, the noise magnitude, the NPS peak and average frequencies were analyzed for each reconstructed CT image set, describing the NPS frequency content [19]. The NPS behavior was studied as a function of reconstruction strength levels for each CTDI_vol_ and for the three different CTDI_vol_ values (1, 5 and 10 mGy).

In order to compare the shape of the NPS curves, the NPS was normalized by the maximum of the FBP NPS curve for each reconstruction strength level using Equation (1):(1)$$NPS_{norm}\left(f\right)=\frac{NPS\;\left(f\right)}{NPS_{FBP}\left(f_{P}\right)}$$

#### 2.2.2. Task Transfer Function

The air rod insert surrounded by the background material was considered relevant to simulate a solid nodule in the lung, as sufficient high contrast was reached for this specific task, Figure 1. The TTF was calculated from the radial edge spread functions passing through the air insert, as explained in the AAPM report [19]. The total CNR must be higher than 15, as recommended in the AAPM report, to achieve a reliable estimate of the TTF.

#### 2.2.3. Detectability Index

The detectability index assessment was based on a clinical task simulated by a non-prewhitening observer model with an eye filter (NPWE). The simulated nodule was a Gaussian spot with a radius of 5 mm. The interpretation conditions were based on the Saunders model with 0.2 mm display pixel pitch, a zoom of 1 and an observer at a distance of 500 mm. The detectability indices were assessed using the NPS and the TTF_air_ of each configuration to evaluate the performance of the PCCT in the case of polyenergetic and virtual monoenergetic reconstructions.

## 3. Results

### 3.1. Acquisition and Reconstruction of the Phantom

The two relevant phantom modules are shown in Figure 1. The homogeneous module of the Catphan phantom gave an HU mean value of 2 within the homogeneous module. No drastic changes in HU within the air insert and the background are presented in Table 2. Consequently, the contrast was considered equivalent for each protocol and well representative of the desired clinical task.

### 3.2. Physical Metrics

#### 3.2.1. Noise Power Spectrum

The NPS curves as a function of the reconstruction levels are shown in Figure 2 for a CTDI_vol_ of 10 mGy for a polyenergetic reconstruction. The NPS shapes were similar for both CTs. For each additional reconstruction strength level, the noise reduction was higher for the PCCT; the first iterative level from WFBP to QIR 1 led to a noise decrease of 29% compared with 14% for the DSCT. Moreover, the normalized NPS curves as a function of the reconstruction levels for the three different CTDI_vol_ are shown in Figure 3. The noise reductions were equivalent at lower CTDI_vol_.

The peak frequency, the averaged frequency and the noise reduction are presented in Table 3 for a CTDI_vol_ of 10 mGy. The peak frequencies were shifted to low spatial frequencies when increasing the reconstruction strength level for both CT scanners. Furthermore, the peak frequencies and averaged frequencies were unchanged for the lower CTDI_vol_.

In the case of the 70 keV reconstruction, the NPS shapes were shifted to lower spatial frequencies for both CT scanners, as shown in Figure 4 and Table 4.

The noise reduction was less efficient for the 70 keV reconstruction, with a total noise reduction of −59% against −71%, but the noise baseline was five times smaller compared with the polyenergetic reconstruction for the PCCT scanner. The peak and averaged frequencies were kept constant when increasing the PCCT reconstruction strength, as they were shifted to lower spatial frequencies for the DSCT, meaning that there are fewer texture changes for PCCT images.

The results found for the 5 and 1 mGy CTDI_vol_ have the same shape as 10 mGy, as seen in Figure 5. However, the noise reduction is a little higher for the highest CTDI_vol_.

#### 3.2.2. Task-Based Transfer Function

The task transfer functions were compared in the case of polyenergetic and virtual monoenergetic acquisitions. As previously said, TTF was calculated for the air insert, reproducing a solid nodule in the lungs. All total CNRs were over 15 (between 19 and 660), as recommended by AAPM [19].

For the reconstruction kernels dedicated to parenchyma, edge enhancement is provided by the algorithm, as seen in Figure 6 [20]. TTF_air_ obtained on both CTs are quite different, where edge enhancement is more pronounced on the PCCT; in fact, the maximum of TTF is higher. Moreover, the spatial resolution is better on DSCT as the f_50%_ is higher, as seen in Table 5. On DSCT, f_50%_ increased with the reconstruction strength level, whereas on the PCCT, the behavior of TTF_air_ was less dependent on it.

For the different CTDI_vol_ acquisitions at 1, 5 and 10 mGy on PCCT, the results were similar for 5 and 10 mGy. For 1 mGy, the results were different, and the enhancement was less pronounced. F_50%_ was higher but still much lower than the results obtained with DSCT, as shown in Table 5.

For the virtual monoenergetic acquisition, TTF_Air_ obtained on both CTs had the same shape, as seen in Figure 4. F_50%_ were a little different at 0.36 vs. 0.40 mm^−1^ for DSCT and PCCT, respectively. In any case, TTF_Air_ seemed independent of the reconstruction strength levels and CTDI_vol_ on each CT, as seen in Table 6.

#### 3.2.3. Detectability Index

The detectability index comparison is illustrated in Figure 7. The detectability index, d’, of the clinical task was calculated based on the previously described parameters. Without IR, d’ was close on both CTs at 162 versus 137 for PCCT and DSCT, respectively, for polyenergetic acquisitions and 265 versus 238 for PCCT and DSCT, respectively, for virtual monoenergetic acquisitions. Nevertheless, d’ increased more sharply on PCCT than on DSCT for high reconstruction strength levels. In fact, d’ for the highest strength levels for DSCT (ADMIRE 5) were 59% and 67% of d’ for PCCT (QIR 4) for polyenergetic and virtual monoenergetic reconstructions, respectively. In terms of d’ values, the highest level of ADMIRE (5) on DSCT was equivalent to QIR 2 on PCCT (264 vs. 267 and 384 vs. 373 for polyenergetic and virtual monoenergetic acquisitions, respectively).

d’ at different CTDI_vol_ were calculated and increased in the same way for all CTDI_vol_, 2.8× and 2.4× between WFBP and QIR 4 for polyenergetic and virtual monoenergetic acquisitions, respectively, in Table 7 and Table 8.

## 4. Discussion

Concerning the detector technology, the PCCT suppresses the electronic noise by defining the first energy threshold just above the electronic noise level (20 keV) [21] but also decreases the noise by suppressing the conversion to light step. However, the noise was of the same magnitude using the polyenergetic reconstruction for both CTs. The PCCT noise using the virtual monoenergetic WFBP reconstruction was higher than that of the DSCT. An explanation could be the generation of the detectors (Stellar^infinity^) present on DSCT, which are already designed to reduce electronic noise [4].

For polyenergetic acquisition, NPSs have the same shape on both CTs. The f_mean_ and the f_peak_ were close, with a maximum difference of 0.04 mm^−1^ between both CTs for an equivalent reconstruction strength level. For the VMI at 70 keV, the NPS shapes between both CTs were different, as shown in Figure 4; nevertheless, the f_mean_ were close, and the maximum difference in the f_peak_ was 0.06 mm^−1^ for the equivalent reconstruction level, as shown in Table 4. The major difference was the shift to low frequencies that is more pronounced for the DSCT. The QIR algorithm seems more constant than ADMIRE, as the shift only appears at the highest reconstruction strength levels with a lower intensity than that of the DSCT.

Sartoretti et al. found similar results for the Bl64 reconstruction kernel, a noise diminution of the same order of magnitude [11]. The noise values for the different QIR levels were 78%, 59%, 43% and 29% of the QIR-off level against 71%, 59%, 43% and 29% in our study. They also observed a shift in f_peak_ for QIR 4 (−6.7%), which is lower than the shifts observed in our study (−12% for QIR 4 and −7% for QIR 3). An explanation could be the sharpness of the kernel used.

Woeltjen et al. studied image quality on patient images rated by physicians on several criteria [10]. They compared images obtained on an energy-integrating detector CT (EID CT) without IR to images obtained on PCCT without IR and with QIR 2. They found that the noise in the PCCT images with IR was lower than that on EID CT in enlarged images of a part of the lung and the same order on axial slices (the whole lung). Moreover, the CTDI_vol_ used on PCCT were lower than on EID CT. These results confirm the contribution of IR in the diminution in noise in images.

Booij et al. obtained close noise and CNR on both DSCT and PCCT, for water and water with iodine inserts in anthropomorphic phantoms, for VMI at 70 keV using the WFBP reconstruction [9]. Similar results were obtained in this study with a global noise of 8.2 HU on both CTs.

The parenchyma reconstruction kernels used in this study provided an edge enhancement. This results in a TTF over 1 for some spatial frequencies. On PCCT, the most enhanced frequency is 0.38 mm^−1^, and on DSCT, it is 0.45 mm^−1^. Moreover, the TTF decreased more rapidly for PCCT (under 0.25 for frequencies over 0.8 mm^−1^), as it was over 0.5 at 0.8 mm^−1^ for all reconstruction strength levels for DSCT. On DSCT, we found TTF behavior similar to a study of Solomon et al. [22], namely, that TTF increased with reconstruction strength levels, but this study was for different contrasts because their reconstruction kernel was suitable for soft tissues.

Moreover, the parenchyma reconstruction kernels used for PCCT were not the ones used in clinical routine for PCCT. Most acquisitions are performed with ultra-high spatial resolution modes using the Bl60 reconstruction kernel, and Sartoretti et al. [11] chose a Bl64 reconstruction kernel to analyze image quality on patient images, which can correspond to clinical routine as well. A study could be conducted on the influence of the reconstruction kernel on ultra-high spatial resolution images obtained with PCCT.

Sartoretti et al. [11] showed that spatial resolution decreased slightly when QIR increased for high contrast with the Bl64 reconstruction kernel. We did not observe a TTF dependence with QIR levels: the TTF shapes were unchanged for QIR 1 to 3, and the WFBP and QIR 4 TTF were superimposed. Their f_peak_ was around 0.4 mm^−1^, which is very close to our result. However, their f_50%_ was higher than ours, reflecting a better spatial resolution. These differences could be explained by the edge spread functions (ESFs) around the insert rod used to compute TTF: Sartoretti et al. based their study on seventy-two radial ESFs with a 10-degree angular aperture, whereas imQuest software was developed from TG-233 recommendations [19] with radial bin widths of 1/10th the image pixel size (0.06 mm). With the 20 cm diameter phantom, the spatial resolution (characterized by TTF) was identical for 5 and 10 mGy, but the edge enhancement was less marked for 1 mGy and complemented by a slight gain for spatial frequencies greater than 0.7 mm^−1^.

An important difference in image acquisition to obtain VMI is the use of only one X-ray tube on PCCT. The second X-ray tube is used for high-pitch or cardiac imaging. Consequently, the acquisition parameters cannot be similar on the two CTs in the VMI. However, these parameters are close to clinical routine, which allows the comparison of the results, as performed by Liu et al. [23]. The TTFs at 70 keV VMI were independent of reconstruction levels on both CTs. The PCCT TTFs were also independent of the value of the CTDI_vol_. The difference in f_50%_ between the two CTs was 0.05 mm^−1^ (0.40 mm^−1^ for PCCT and 0.35 mm^−1^ for DSCT). The VMI spatial resolution has not been studied so far using TTF [9,24,25]. One limitation of our study was that the TTF was analyzed for only one VMI at 70 keV. It would be very interesting to know if this independence of the TTF from the reconstruction kernel and dose level appeared for other VMI energies.

In our acquisition conditions, the TTFs were little influenced by reconstruction strength levels and dose levels using the polyenergetic reconstruction. The TTFs using the VMI reconstruction were unchanged, whatever the reconstruction strength and dose levels for both CT scanners. The detectability index, therefore, increased proportionally with the noise diminution due to the increase in the reconstruction strength level and dose. As expected, the highest detectability index was obtained on PCCT for 10 mGy at QIR 4. Sartoretti et al. showed a spatial resolution reduction with a reconstruction strength level increase [11]. They concluded that QIR 3 was the best compromise strength between noise reduction and spatial resolution.

Si-Mohamed et al. investigated the differences in image quality and detectability between various CT scanners and algorithms for Philips [14,16]. A smaller noise gap between the DSCT and the PCCT of the Siemens company was observed compared to the Philips CT. Furthermore, the Philips PCCT images produced a ring artifact, which was highlighted at low NPS frequencies. The detectability index increased much faster for PCCT than DSCT on the Siemens CT scanners, which is in contrast to the results obtained for the Philips preclinical prototype [14]. However, Si-Mohamed et al. and our study found a PCCT detectability increase compared to energy-integrating detector CTs for solid lung nodules, improving the clinical routine.

Bartlett et al. [15], who compared patient images on energy-integrated detector CT and PCCT for WFBP, found a better nodule detection for only one reader of two for PCCT images. Our WFBP results are in accordance with close detectability indices: 137 and 162 for DSCT and PCCT, respectively, for polyenergetic acquisition.

The detectability indices calculated in our study are also higher than those in a study of Si-Mohamed et al. for the same contrast. This could be explained, in part, by a different clinical task, as we took a nodule of 5 mm diameter against 4 mm [14].

Another limitation of our study came from the absence of scoring images by a radiologist. Even if the detectability index aims to reproduce the radiologist’s eyes, it does not consider the radiologist’s feeling in front of the image. It was proven in a previous study that even though the detectability index increased and noise decreased, radiologists do not rate the image quality as better [11,14]. Those images were qualified as “artificial”. Moreover, only one energy VMI was studied. A further study could provide different VMI energies to characterize the benefit of the different reconstructions to diagnostic tasks.

## 5. Conclusions

The PCCT outperformed the DSCT with a better detectability index of the solid nodule in the lung. The virtual monoenergetic reconstruction showed its superiority in terms of noise level before running the iterative reconstruction and in terms of TTF.

## Figures and Tables

**Figure 1 diagnostics-13-01325-f001:**
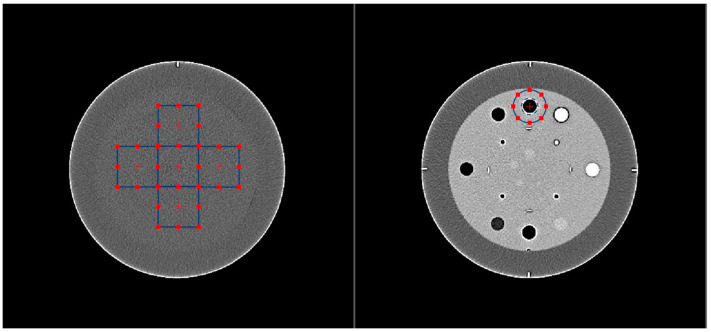
Visualization of the two relevant modules of the CATPHAN phantom and imQuest ROI for physical metrics evaluation. The left picture shows the homogeneous module, and the right picture is the module with the inserts.

**Figure 2 diagnostics-13-01325-f002:**
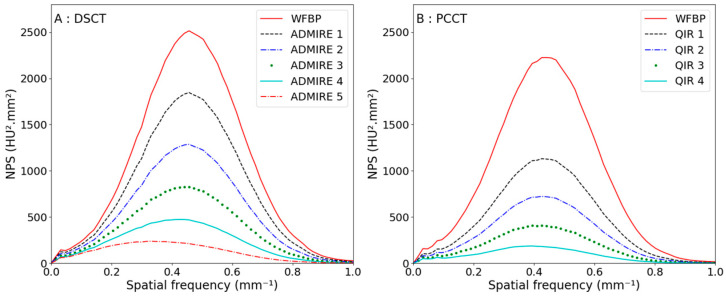
Noise power spectra curves using polyenergetic reconstruction with a CTDI_vol_ of 10 mGy when increasing the iteration level of the reconstruction algorithm for the DSCT (**A**) and the PCCT (**B**).

**Figure 3 diagnostics-13-01325-f003:**
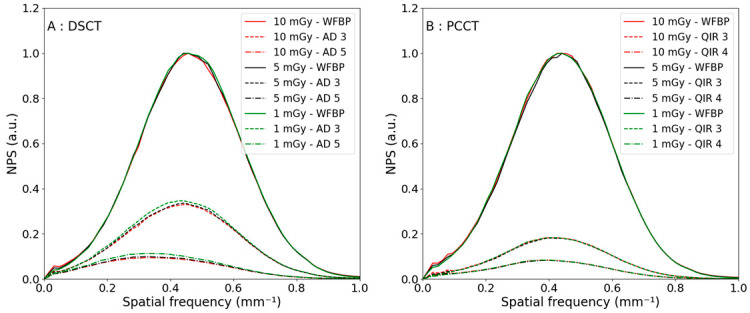
Noise power spectra curves in the case of polyenergetic reconstruction, with a CTDI_vol_ of 10 mGy in red, 5 mGy in black and 1 mGy in green, when increasing the iteration level of the reconstruction algorithm for the DSCT (**A**) and the PCCT (**B**). WFBP—weighted filtered back projection and AD—ADMIRE.

**Figure 4 diagnostics-13-01325-f004:**
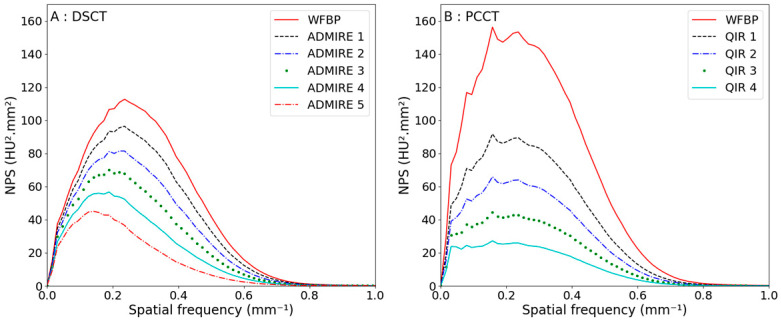
Noise power spectra curves in the case of monoenergetic images for a CTDI_vol_ of 10 mGy when increasing the reconstruction level for the DSCT (**A**) and the PCCT (**B**).

**Figure 5 diagnostics-13-01325-f005:**
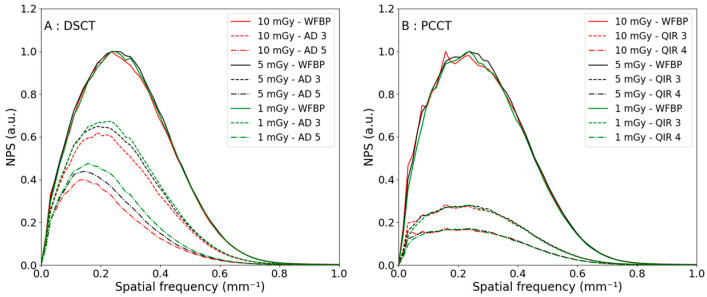
Noise power spectra curves in the case of virtual monoenergetic reconstruction, with a CTDI_vol_ of 10 mGy in red, 5 mGy in black and 1 mGy in green, when increasing the iteration level of the reconstruction algorithm for the DSCT (**A**) and the PCCT (**B**). WFBP—weighted filtered back projection and AD—ADMIRE.

**Figure 6 diagnostics-13-01325-f006:**
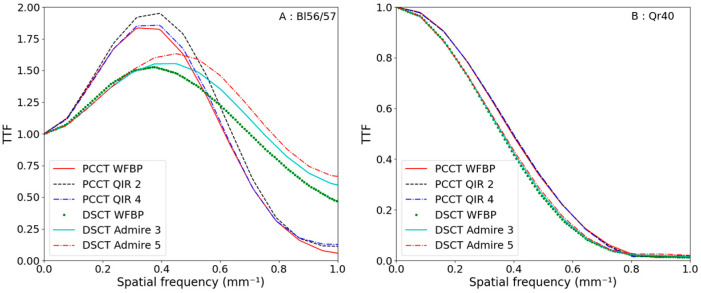
TTF_Air_ obtained on both CTs for polyenergetic and virtual monoenergetic acquisitions for 3 reconstruction strength levels at 10 mGy.

**Figure 7 diagnostics-13-01325-f007:**
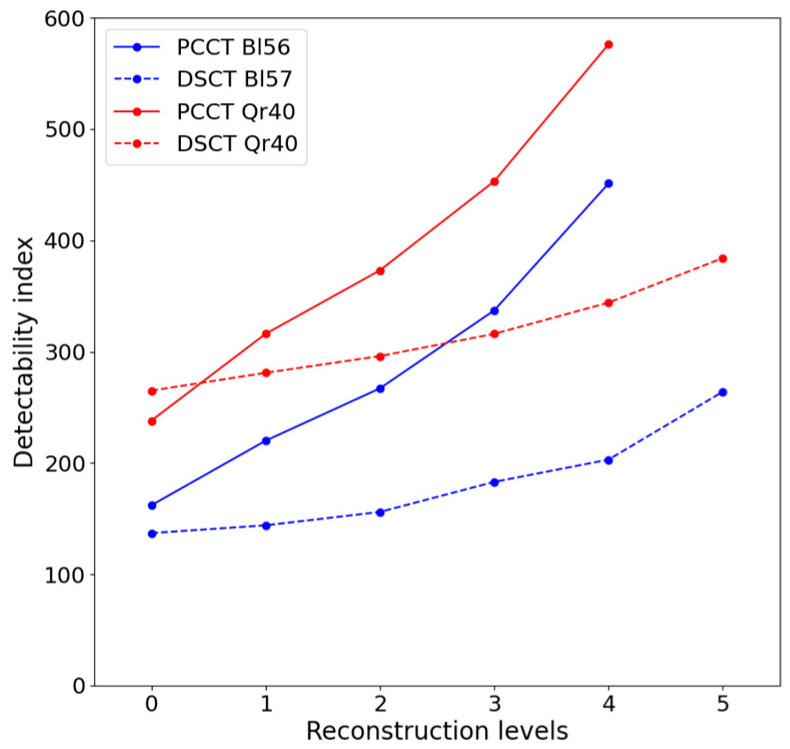
Detectability index on both CTs for polyenergetic (Bl56 and Bl57) and virtual monoenergetic (Qr40) acquisitions and for all reconstruction strength levels.

**Table 1 diagnostics-13-01325-t001:** Acquisition and reconstruction parameters for the PCCT and DSCT scanners.

	Parenchyma		VMI 70 keV	
	DSCT (1) *	PCCT (2) *	DSCT (3) *	PCCT (4) *
Tube	A	A	A + B	A
High voltage (kV)	120	120	80/Sn 150	120
Collimation (mm)	57.6	57.6	57.6	57.6
Pitch	1.5	1.5	1	1
Exposure time per rotation (s)	0.5	0.5	0.25	0.25
Field of view (mm)	300	300	300	300
Matrix size (pixels)	512 × 512	512 × 512	512 × 512	512 × 512
Reconstruction strength levels	WFBP + ADMIRE 1 to 5	WFBP + QIR 1 to 4	WFBP + ADMIRE 1 to 5	WFBP + QIR 1 to 4

* acquisition number. DSCT—dual-source CT; PCCT—photon-counting CT; VMI—virtual monoenergetic imaging; WFBP—weighted filtered back projection; QIR—quantum iterative reconstruction; ADMIRE—Advanced Modeled Iterative Reconstruction.

**Table 2 diagnostics-13-01325-t002:** HU mean values in the air insert and in the background were measured for the CTDI_vol_ of 10 mGy. The acquisition number correspondence is indicated in Table 1.

Acquisition	1	2	3	4
HU (air)	−983	−1042	−968	−1007
HU (background)	98	94	95	97
Contrast	1081	1136	1063	1104

**Table 3 diagnostics-13-01325-t003:** Global noise and its noise reduction in percentages relative to WFBP, peak frequency (f_peak_) and averaged frequency (f_mean_) values for all reconstruction levels for both CTs.

	DSCT			PCCT		
	Noise (HU)	f_peak_ (mm^−1^)	f_mean_ (mm^−1^)	Noise (HU)	f_peak_ (mm^−1^)	f_mean_ (mm^−1^)
WFBP	54.8	0.46	0.47	50.0	0.44	0.44
1	47.1 (−14.1%) *	0.46	0.46	35.7 (−28.6%) *	0.43	0.43
2	39.4 (−28.1%) *	0.46	0.45	28.5 (−43.0%) *	0.43	0.43
3	31.8 (−42.0%) *	0.44	0.44	21.4 (−57.2%) *	0.41	0.42
4	24.2 (−55.8%) *	0.43	0.42	14.4 (−71.2%) *	0.39	0.4
5	16.8 (−69.3%) *	0.35	0.38	-	-	-

*: noise reduction in percentages relative to WFBP.

**Table 4 diagnostics-13-01325-t004:** Global noise and its noise reduction in percentages relative to WFBP, peak frequency (f_peak_) and averaged frequency (f_mean_) values for all reconstruction levels for both CTs for a 10 mGy acquisition.

	DSCT			PCCT		
	Noise (HU)	f_peak_ (mm^−1^)	f_mean_ (mm^−1^)	Noise (HU)	f_peak_ (mm^−1^)	f_mean_ (mm^−1^)
WFBP	9.0	0.25	0.29	10.7	0.19	0.28
1	8.2 (−8.9%) *	0.22	0.28	8.2 (−23.4%) *	0.19	0.28
2	7.4 (−17.8%) *	0.22	0.27	6.9 (−35.5%) *	0.19	0.27
3	6.6 (−26.7%) *	0.20	0.26	5.6 (−47.7%) *	0.19	0.27
4	5.6 (−37.8%) *	0.17	0.25	4.4 (−58.9%) *	0.19	0.26
5	4.5 (−50.0%) *	0.16	0.23	-	-	-

*: noise reduction in percentages relative to WFBP.

**Table 5 diagnostics-13-01325-t005:** F_50%_ (mm^−1^) of PCCT and DSCT for polyenergetic acquisitions for all reconstruction strength levels and for the three CTDI_vol_ (1, 5, 10 mGy).

	DSCT-Bl57			PCCT-Bl56		
	10 mGy	5 mGy	1 mGy	10 mGy	5 mGy	1 mGy
WFBP	0.97	0.96	0.91	0.73	0.75	0.75
1	1.13	1.06	1.17	0.75	0.76	0.77
2	1.23	1.18	1.24	0.75	0.76	0.79
3	1.29	1.25	1.28	0.75	0.77	0.80
4	1.40	1.39	1.32	0.73	0.76	0.82
5	1.49	1.48	1.34	-	-	-

**Table 6 diagnostics-13-01325-t006:** F_50%_ (mm^−1^) of PCCT and DSCT for virtual monoenergetic acquisition for all reconstruction strength levels and for the three CTDI_vol_ (1, 5, 10 mGy).

	DSCT			PCCT		
	10 mGy	5 mGy	1 mGy	10 mGy	5 mGy	1 mGy
WFBP	0.35	0.35	0.35	0.39	0.39	0.39
1	0.36	0.35	0.35	0.40	0.39	0.39
2	0.36	0.35	0.35	0.40	0.39	0.39
3	0.36	0.35	0.35	0.40	0.40	0.40
4	0.36	0.35	0.35	0.40	0.40	0.40
5	0.36	0.36	0.35	-	-	-

**Table 7 diagnostics-13-01325-t007:** Detectability index values when varying the IR level and the CTDI_vol_ for the PCCT scanner for polyenergetic acquisition.

Reconstruction Strength Level	10 mGy	5 mGy	1 mGy
0	162	110	49
1	220	153	67
2	267	184	82
3	337	233	103
4	451	311	140

**Table 8 diagnostics-13-01325-t008:** Detectability index values when varying the IR level and the CTDI_vol_ for the PCCT scanner for virtual monoenergetic acquisition.

Reconstruction Strength Level	10 mGy	5 mGy	1 mGy
0	238	175	81
1	316	227	106
2	373	268	124
3	453	325	151
4	576	413	192

## Data Availability

Not applicable.

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
