# Peer review of "Image Quality Analysis of Photon-Counting CT Compared with Dual-Source CT: A Phantom Study for Chest CT Examinations"

_diagnostics, 2023, doi:10.3390/diagnostics13071325_

Round 1

Reviewer 1 Report

Review - Image quality analysis of photon-counting CT compared with dual-source CT: a phantom study for chest CT examinations

The provided manuscript by Deleu et al. deals with the very interesting topic of image quality of PCCT compared to DECT with an EI detector. This is a noteworthy article. However, some clarifications and amendments should be made.

Title:

The title is adequate.

Abstract:

The statement about superiority in lung nodule detection is not included in the abstract.

Introduction:

p.1 LL 27-28: “in other” please explain or rephrase.

p.1 LL 39-40: the authors are probably right, however this will have to be proven true within the coming years. Please consider rephrasing.

p.1 LL85-86: “Both systems CTDIvol displayed were controlled before doing the 85 measurements and were in conformity with national regulation.” Please explain in more detail, what was controlled and how? What national regulation?

Methods:

p.2 LL75-76: VA40 and VB20 are software version numbers, please indicate within the text.

p.2 LL83-84: which national reference levels? please provide reference.

Results:

Table 4,5 and 6 are not completely visible within the .pdf file. Please check.

Discussion:

p.9 LL. 183-185: The sentence is a bit odd. Please consider rephrasing it (maybe two sentences) to make the point clearer.

p.10 LL 190-196: Comparisons with other manufacturers seems very interesting. However, the paragraph seems a bit out of place here. Please consider an introductory sentence to make it clearer to the reader.

p.10 LL. 221-228: In clinical routine, we use different kernels when reconstructing PCCT images compared to conventional CT, especially when using optimal spatial resolution (0.2 mm). In my opinion, it would be interesting to include different kernels... perhaps in a future study.

Reviewer 2 Report

This innovative technique is interesting: could you give some details on its possible clinical application, eventually supplemented by some images?
